# Direct visualization of a molecular handshake that governs kin recognition and tissue formation in myxobacteria

Pengbo Cao[1,2] & Daniel Wall[1]

Many organisms regulate their social life through kin recognition, but the underlying mechanisms are poorly understood. Here, we use a social bacterium, *Myxococcus xanthus*, to investigate kin recognition at the molecular level. By direct visualization of a cell surface receptor, TraA, we show how these myxobacteria identify kin and transition towards multicellularity. TraA is fluid on the cell surface, and homotypic interactions between TraA from juxtaposed cells trigger the receptors to coalesce, representing a 'molecular handshake'. Polymorphisms within TraA govern social recognition such that receptors cluster only between individuals bearing compatible alleles. TraA clusters, which resemble eukaryotic gap junctions, direct the robust exchange of cellular goods that allows heterogeneous populations to transition towards homeostasis. This work provides a conceptual framework for how microbes use a fluid outer membrane receptor to recognize and assemble kin cells into a cooperative multicellular community that resembles a tissue.

[1] Department of Molecular Biology, University of Wyoming, 1000 E University Avenue, Laramie, WY 82071, USA. [2] Present address: School of Biological Sciences, Georgia Institute of Technology, Atlanta, GA 30332, USA. Correspondence and requests for materials should be addressed to D.W. (email: dwall2@uwyo.edu)

K in recognition exists in a wide range of taxa. The ability to recognize kin allows related individuals to form social groups, such as colonies of ants, schools of fish, flocks of birds, and herds of mammals. Social groups, in turn, provide fitness benefits with respect to predation, defense, and raising offspring, among many others. Although understanding the molecular mechanisms underlying kin recognition is a fundamental biological question, such efforts are blocked by the inherent complexity of animal systems, which typically involve the five senses and cognition. Recently single-celled microbes were shown to recognize kin to direct various multicellular activities[1,2], and therefore, they provide attractive and tractable platforms to investigate kin recognition and the emergence of sociality and multicellularity[3–9].

Microbes produce perceptible cues that reflect their self-identity to recognize proximal individuals bearing the same cues. Recognition, in turn, facilitates related cells to engage in cooperative behaviors. Strikingly, some microbes are able to transition into a state where related individuals assemble themselves into an organized multicellular entity. Such transitions allow cells to differentiate into different types with specialized functions and to coordinate activities that are beyond the capabilities of individuals[1,10,11]. Understanding how cells recognize kin and make lifestyle transitions can provide valuable insights into the emergence of complex multicellular life.

*Myxococcus xanthus* is a soil bacterium that is well known for its social behaviors and its ability to transition from unicellular into multicellular life by an aggregation strategy. We previously discovered two outer membrane (OM) proteins in *M. xanthus*, named TraA and TraB, that function together in cell–cell recognition and adhesion[3,8]. TraA is a highly polymorphic cell surface receptor that recognizes related individuals bearing identical or nearly identical receptors by homotypic interactions[3,12]. Selectivity in recognition is mediated by the variable domain (VD) of TraA[3,8]. Upon TraA–TraA recognition, cells undergo a process termed OM exchange (OME) during which the bulk of OM proteins and lipids are bidirectionally transferred between individuals[13–15]. TraB contains a predicted β-barrel domain and an OmpA cell wall binding domain and is required for TraA to form a functional adhesin for OME[8,13]. TraA is secreted and thought to be anchored on the cell surface by its MYXO-CTERM sorting tag[13]. Because lipids are also transferred, OME is thought to be driven by membrane fusion[13,16,17], which, in turn, facilitates diverse social interactions including communication, cellular repair, and kin discrimination[18–20]. Thus, TraA/B play key roles in modulating the sociality of myxobacteria. However, the molecular mechanisms underlying how cells use this adhesin (a hypothetical fusogen) to recognize kin and transition towards a multicellular lifestyle remain largely unexplored.

Here, we set out to visualize the molecular dynamics of TraA/B during OME and during the transition toward multicellularity. We show that, upon physical contact, sibling cells bearing compatible TraA receptors undergo a molecular 'handshake.' That is, the receptors from juxtaposed cells rapidly coalesce into clusters at the sites of cell–cell contact. We consider that the fluid nature of these OM proteins facilitates myxobacteria transition between unicellular and multicellular lifestyles. In addition, we show that the formation of TraA clusters was followed by the rapid transfer of cellular goods between cells. The robust exchange of cell contents among kin allows distinct and heterogeneous populations to transition toward uniformity that resembles a tissue.

## Results

**TraA is a fluid receptor that clusters at cell–cell contacts**. To directly observe the dynamics of TraA during OME, we fused the receptor with fluorescent proteins (FPs). Because TraA is processed at both its N and C termini during its transport to the cell surface[8,13], we placed a superfolder green fluorescent protein (GFP) in the middle of TraA where two dispensable cysteine-rich repeats (C2 and C3)[12] are located and obtained a functional fusion (TraA-GFP; Fig. 1 and Supplementary Fig. 1). To allow a

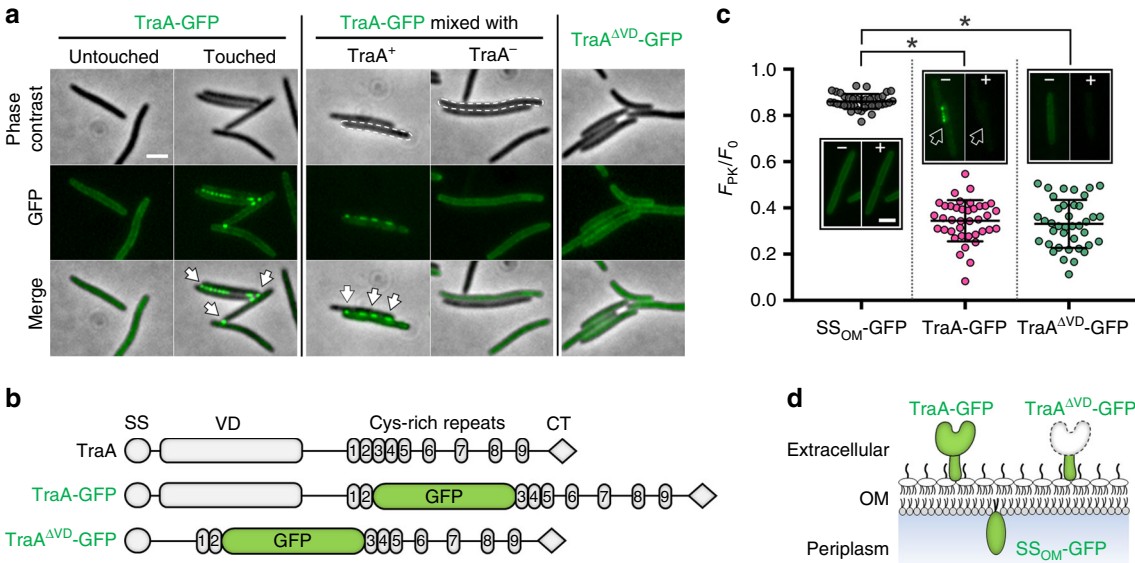

**Fig. 1** TraA receptors cluster at cell–cell contact interfaces. **a** The formation of TraA-GFP foci (arrows) requires direct contacts between *traA*⁺ cells. **b** Domain architectures of TraA and constructed variants used in (**a**). SS, type I signal sequence; VD, variable domain; CT, MYXO-CTERM. **c** Cell surface localization of TraA fusion proteins revealed by PK accessibility assay. Arrows highlight the disappearance of TraA-GFP foci after PK treatment. Effects were quantified by comparing fluorescence of PK-treated cells relative to that of the same cells before treatment. $SS_{OM}$-GFP is a control localized on the periplasmic side of the OM. $F_{PK}$ (fluorescence intensity of whole cells after PK treatment); $F_0$ (fluorescence intensity before PK treatment). Representative images of cells before (−) and after (+) PK treatment are shown. Error bars represent standard deviation from the mean ($n = 39$ in each case). Significant differences are indicated with asterisks ($p < 0.05$; $t$-test). **d** Schematic of localization results from (**c**). See Supplementary Table 1 for strain details. Scale bar = 1 μm. Source data are provided as a Source Data file

clear fluorescence signal, TraA-GFP was expressed from the chromosome by the heterologous *pilA* promoter in a *M. xanthus* strain lacking its endogenous *traA*. Visualization of this strain revealed a uniform distribution of the TraA-GFP signal throughout the cell envelope when cells were isolated (Fig. 1a). Remarkably, when cells made physical contact, discrete TraA-GFP foci formed at cell–cell junctions. Foci formation required the presence of TraA in both cells, as contacts between TraA-GFP and Δ*traA* cells did not produce foci, suggesting that TraA recognition between adjacent cells triggers the receptors to coalesce. When live cells were treated with proteinase K (PK), the TraA-GFP signal disappeared, indicating that the fusion protein was surface exposed (Fig. 1c, d). To verify that the GFP reporter did not trigger foci formation, a fusion was created whereby the recognition domain, VD, was deleted (TraA$^{\Delta VD}$-GFP; Fig. 1b). This fusion protein was properly sorted to the cell surface, but it

no longer clustered at cell–cell contacts (Fig. 1), indicating that the VD was required for foci formation.

The ability of TraA to form clusters suggested that it is fluid on the OM. Using time-lapse microscopy, we monitored the interactions between two motile cells expressing functional TraA-mCherry fusions (Fig. 2a and Supplementary Fig. 1, Supplementary Movie 1). Notably, TraA-mCherry formed foci in ≤30 s upon cell–cell contact, and those foci rapidly disappeared after cell separation, indicating that TraA dynamics were affected by cell–cell contacts. Fluorescence recovery after photobleaching (FRAP) was used to further assess TraA fluidity. After a portion of a cell was photobleached, the fluorescence of TraA-GFP was readily recovered, demonstrating that TraA was indeed fluid on the cell surface (Fig. 2b and Supplementary Fig. 2). In an alternative approach, when isolated live cells were treated with antibodies that target the VD, TraA-GFP foci formed upon

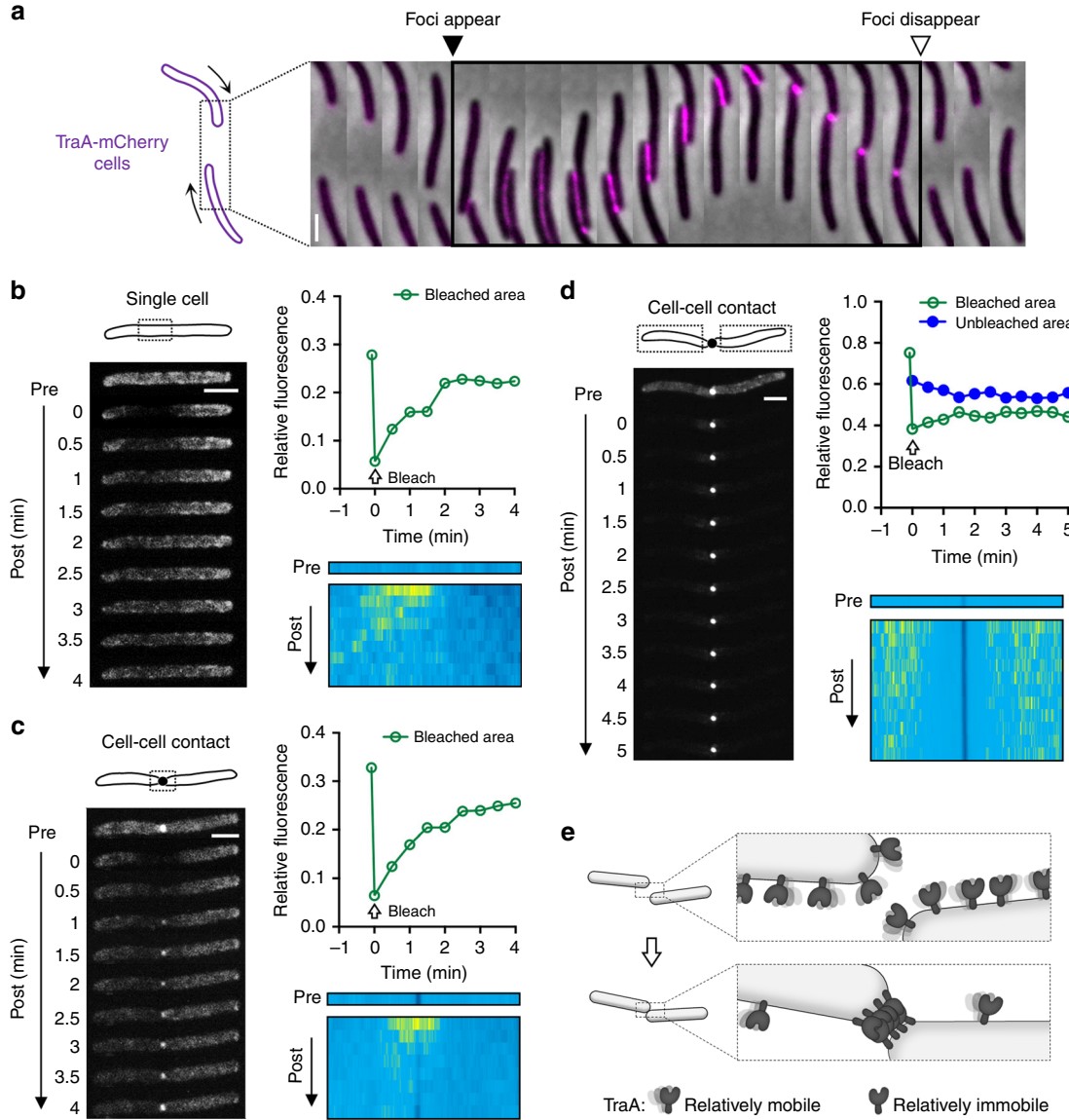

**Fig. 2** Dynamics of TraA receptors at the cell surface. **a** A representative time series of interactions between two cells bearing TraA-mCherry. Time interval between frames was 30 s. **b**–**d** FRAP analyses of TraA-GFP fluidity on the cell surface. Representative FRAP images are shown. Relative fluorescence measures the ratio of fluorescence intensity of the bleached areas (indicated with dashed borders) to that of the whole cells before and after photobleaching. TraA-GFP fluorescence intensity profiles along the long axes of analyzed cell(s) before and after photobleaching are shown as kymographs. Blue, high intensity; yellow, low intensity. **e** A model illustrating how TraA mobility changes upon cell–cell contact. Scale bar = 1 μm. Source data are provided as a Source Data file

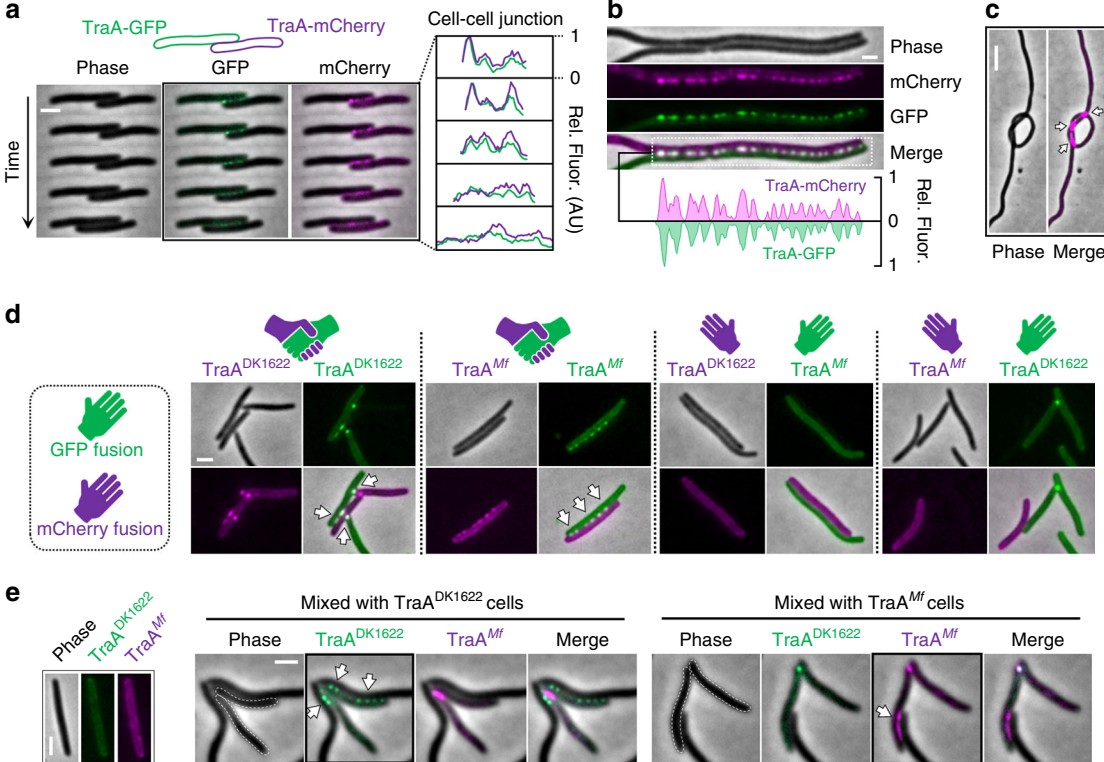

**Fig. 3** A molecular handshake between TraA receptors governs cell–cell recognition. **a** A representative time series (45 s between frames) shows dynamic handshakes between a TraA-GFP cell and a TraA-mCherry cell. Fluorescence intensities at cell–cell contacts are quantified. Fluorescence was measured in arbitrary units (AU) throughout this study. **b** Representative images showing the colocalization of TraA-GFP and TraA-mCherry foci between two filamentous cells treated with cephalexin. Fluorescence intensity profiles along the entire cell–cell junction are shown below. **c** Self-contact of a filamentous TraA-mCherry cell caused foci formation. A single cell that tied a knot is shown (see Supplementary Fig. 5 for more examples). **d** TraA receptors cluster and colocalize (indicated with arrows) between cells harboring compatible traA alleles, whereas no clusters form between cells with incompatible alleles, i.e., DK1622 + Mf. **e** A merodiploid strain (indicated with dashed outlines) expressing both TraA$^{DK1622}$-mCherry and TraA$^{Mf}$-GFP selectively clusters its different receptors when encountering cognate unlabeled cells. Black borders and white arrows highlight specific recognition between the merodiploid strain and unlabeled social partners. Images shown in (**c**, **d**) represent the experimental observation of >100 cell–cell contacts in each mixing combination. See Supplementary Table 1 for strain details. Scale bar = 1 μm. Source data are provided as a Source Data file

multivalent antibody binding (Supplementary Fig. 3), again supporting that TraA is diffusible on the cell surface. The observation of foci also indicates that TraA mobility was restricted within clusters upon antibody binding. FRAP was then used to examine the clustering of TraA at cell–cell contacts. After photobleaching the region of intercellular contact (Fig. 2c and Supplementary Fig. 2), the fluorescent foci readily recovered, indicating that unbleached TraA-GFP molecules entered and accumulated at the adhesion site. To test whether TraA within clusters displayed altered fluidity, portions of cells that were not in contact were photobleached, and fluorescence of these regions did not substantially recover (Fig. 2d and Supplementary Fig. 2). In contrast, the foci at the cell–cell contact retained their fluorescence over time. Based on these and prior findings[8], we suggest that homotypic TraA–TraA binding at cell–cell contacts constrains the mobility of TraA receptors and allows them to form clusters (Fig. 2e).

**A molecular handshake governs kin recognition in myxobacteria.** To directly test for TraA homotypic interactions, we used time-lapse microscopy to monitor the interactions between two strains: one bearing TraA-mCherry and the other bearing TraA-GFP. Here TraA-mCherry and TraA-GFP foci appeared simultaneously upon cell–cell contacts (Supplementary Fig. 4). As shown in Fig. 3a, although the fluorescence patterns at the cellular

interface changed dynamically when two cells physically interacted, the TraA-mCherry and TraA-GFP signals were always colocalized, supporting a homotypic interaction model. This experiment was repeated with cells that were first treated with cephalexin to induce filamentation. Again, under these conditions, the fluorescence of TraA-mCherry and TraA-GFP colocalized along the entire cell interface (Fig. 3b). Interestingly, foci also formed when single filamentous cells looped back and touched themselves (Fig. 3c and Supplementary Fig. 5), indicating that interactions between TraA receptors from opposing membranes, and not necessarily between distinct cells, drives foci formation. As noted above, TraA also forms clusters in single cells upon multivalent antibody binding (Supplementary Fig. 3), supporting a model that TraA–TraA binding between cells drives foci formation.

Sequence polymorphisms within TraA determine the selectivity in cell–cell adhesion and OME[3,8]. Here we sought to directly visualize the interactions between isogenic cells expressing different receptors. To this end, FPs were fused to TraA from *Myxococcus fulvus* HW-1 (*Mf*), a receptor that does not recognize TraA from *M. xanthus* DK1622[3,8], which was used in the preceding assays. TraA$^{Mf}$-FP fusions were functional (Supplementary Fig. 1). As expected, the mixing of TraA$^{Mf}$-mCherry and TraA$^{Mf}$-GFP strains led to colocalization of TraA foci at cell–cell contacts (Fig. 3d). Strains expressing TraA$^{Mf}$-FPs or TraA$^{DK1622}$-FPs were then mixed. Notably, TraA did not coalesce between

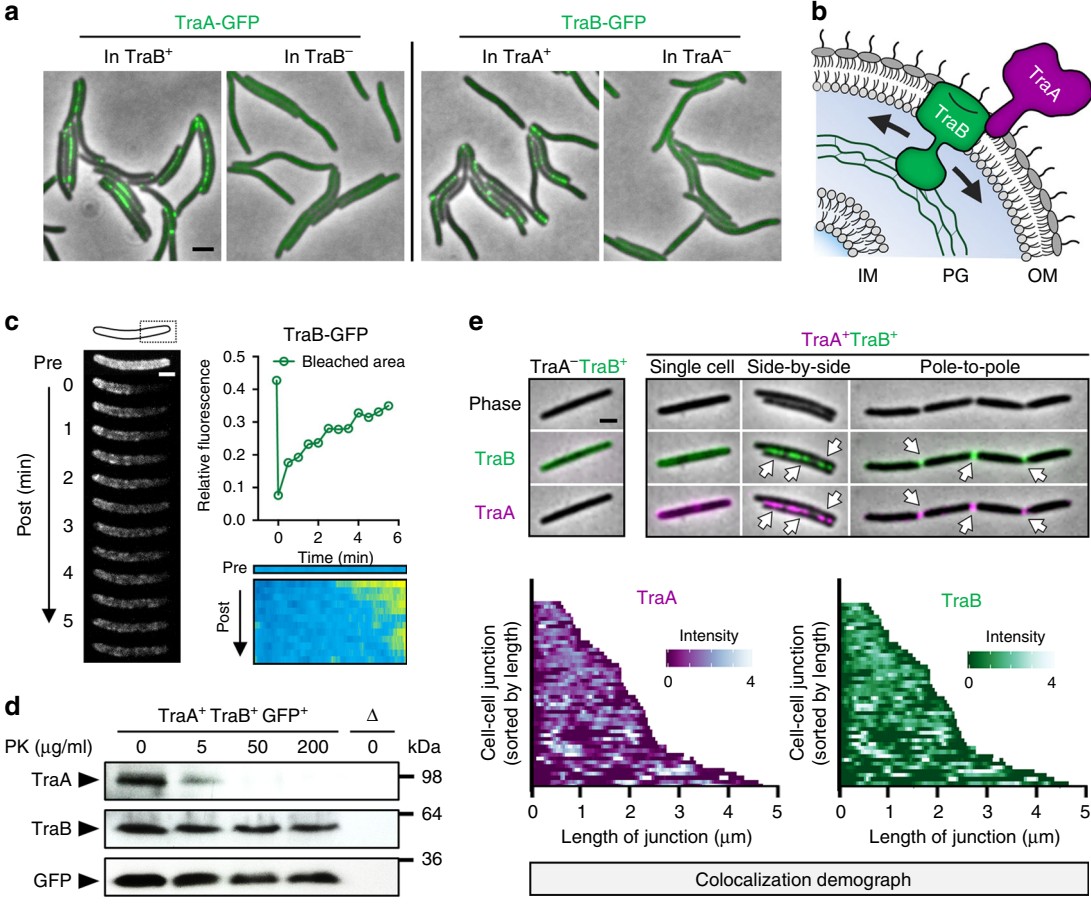

**Fig. 4** TraA and TraB form dynamic adhesins at intercellular junctions. **a** Clustering of TraA or TraB at cell–cell contacts requires the presence of both proteins. Scale bar = 1 μm. **b** A schematic of the TraA/B adhesin complex in the cell envelope. To explain its mobility, TraB likely slides along or is transiently bound to peptidoglycan (PG). IM, inner membrane. **c** Representative FRAP analysis of TraB-GFP fluidity in the OM done as described in Fig. 2 legend. Scale bar = 0.5 μm. **d** Immunoblot analysis of a strain expressing TraA, TraB, and SS_OM-GFP treated with different concentrations of PK. The same samples were probed with anti-TraA serum, anti-TraB serum, or anti-GFP antibody. A strain lacking TraA, TraB, and SS_OM-GFP (indicated as an open triangle) was used as a negative control. **e** Colocalization of TraA and TraB at contact interfaces. TraA was labeled with primary TraA antibodies and a secondary Alexa Fluor 594–conjugated antibody. A TraA⁻TraB⁺ strain was used as a negative control for TraA immunofluorescence (left). TraB was tagged with GFP. Demographs display a collection of 48 fluorescence profiles of TraA and TraB at cell–cell contact interfaces, sorted from the shortest to the longest junctions. Scale bar = 1 μm. Strain details are in Supplementary Table 1. Source data are provided as a Source Data file

cells bearing incompatible TraA receptors; foci only formed between compatible TraA-bearers (Fig. 3d). In addition, a merodiploid strain expressing two different receptors, TraA^Mf-mCherry and TraA^DK1622-GFP, was made. This dual-labeled strain was mixed with non-labeled strains containing either TraA^DK1622 or TraA^Mf, to test whether the merodiploid would selectively use its receptors in recognition. Strikingly, when mixed with unlabeled TraA^DK1622 cells, TraA^DK1622-GFP clustered at the sites between merodiploid cells and TraA^DK1622 cells, whereas TraA^Mf-mCherry foci only formed at the interface between two merodiploid cells (Fig. 3e). Consistent with this, the opposite foci distribution was observed when the merodiploid was mixed with unlabeled TraA^Mf cells (Fig. 3e). We conclude that TraA undergoes an allele-specific molecular handshake to identify kin cells bearing compatible receptors.

**TraA/B is a dynamic adhesin that governs cellular exchange.** TraB assists TraA to form functional adhesins[8]. Consistent with this, in a ΔtraB background, TraA-GFP was uniformly distributed on cells and did not form clusters at cell–cell contacts (Fig. 4a). Sequence analysis suggests that TraB contains an OM-embedded β-barrel domain and a cell wall binding OmpA domain (Supplementary Fig. 6A). To elucidate the dynamics of TraB during OME, a functional fusion was constructed (TraB-GFP; Supplementary Fig. 6A–C). The OM localization of TraB-GFP was confirmed by a plasmolysis experiment (Supplementary Fig. 6D). In support of our adhesion model, TraB-GFP also clustered at cell–cell contacts, and the formation of these foci was dependent on TraA (Fig. 4a). Given that TraB contains domains embedded in the OM and bound to the cell wall (Fig. 4b), along with reports that the OM of Gram-negative bacteria (i.e., Escherichia coli) is rigid[21–23], it was striking that TraB-GFP was fluid in M. xanthus (Fig. 4c). In contrast to TraA, TraB in live cells was resistant to PK treatment (Fig. 4d and Supplementary Fig. 6E), suggesting that it has minimal cell surface exposure. Based on this and other findings[8], we propose that TraB is not directly involved in cell–cell recognition, although it associates with TraA in the OM to function as an adhesin. In support of this, TraB clustered between cells bearing matching TraA, but not between cells bearing incompatible receptors (Supplementary Fig. 7). Furthermore, TraA receptors labeled by immunofluorescence colocalized with TraB-GFP at intercellular junctions (Fig. 4e), suggesting that these fluid OM proteins tightly associate with one another within clusters (Fig. 4b).

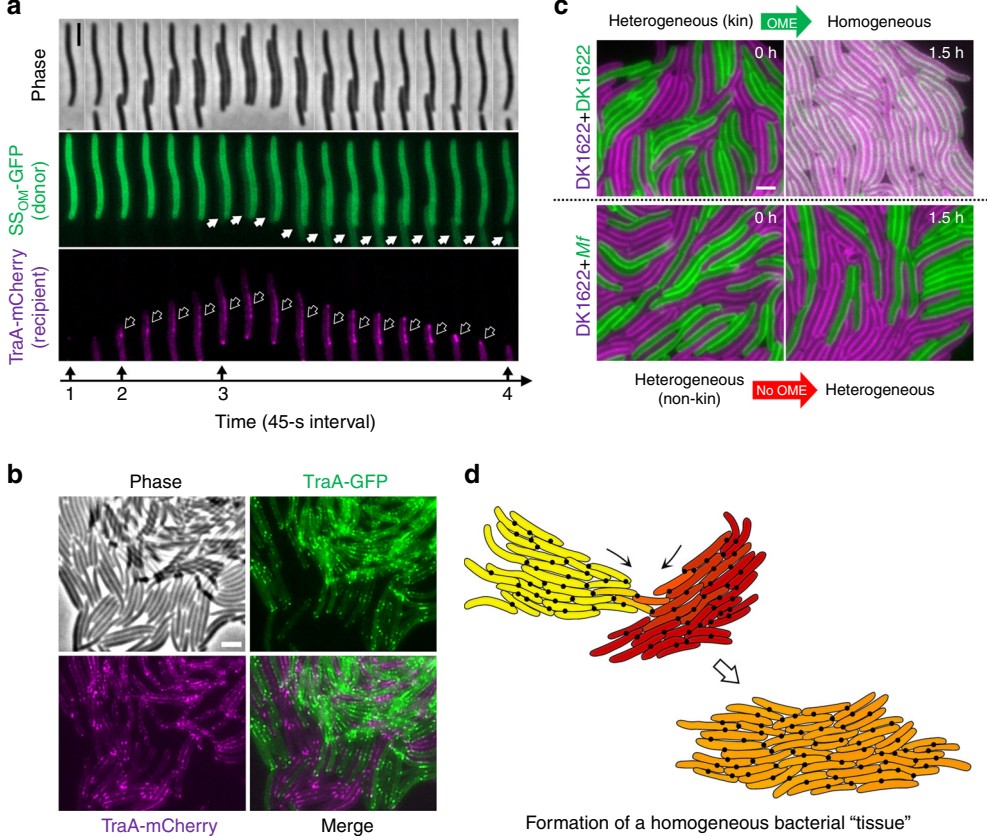

**Fig. 5** TraA/B adhesins transition bacterial kin cells into a tissue. **a** Representative time series showing TraA recognition followed by the transfer of SS$_{OM}$-GFP between cells. Marked in the timeline: 1, before contact no foci are detected; 2, upon contact the TraA foci form (hollow arrows); 3, cargo transfer is detected (solid arrows); 4, cell separation and dissolution of TraA foci. **b** TraA junctions assemble bacteria into a tissue-like structure. TraA-GFP cells were co-incubated with TraA-mCherry cells at a high density (a sub-view of images shown in Supplementary Fig. 11). **c** TraA recognition governs the bidirectional transfer between cells harboring SS$_{OM}$-GFP or SS$_{OM}$-mCherry cellular cargo. Compatible TraA receptors allow exchange after the indicated incubation time (top panels), whereas incompatible receptors block exchange (bottom panels). *traA* alleles shown at the left. See Supplementary Fig. 12 for a more detailed analysis. Strain details are in Supplementary Table 1. Scale bar = 1 μm. **d** A schematic illustration of two kin populations that are phenotypically divergent (labeled yellow and red) that recognize and merge into a homogeneous tissue-like population through OME (orange). The extracellular matrix (polysaccharides and proteins, not shown here) also contribute to tissue formation. Black dots represent TraA/B foci located at cell–cell contacts

TraA recognition leads to the exchange of cellular goods. Here we directly observed this process by mixing a donor strain expressing fluorescent cargo (SS$_{OM}$-GFP) with a recipient expressing TraA-mCherry. As expected, TraA-mCherry formed foci upon cell–cell contact, and these foci disappeared when the cells separated by gliding motility (Fig. 5a). Notably, the recipient efficiently acquired fluorescent cargo from the donor shortly after cell–cell contact occurred and TraA foci formed (Fig. 5a, and Supplementary Movie 2). As a negative control, the same recipient was mixed with a Δ*traAB* donor expressing SS$_{OM}$-GFP, and neither TraA foci nor cargo transfer was observed (Supplementary Fig. 8). Although OME was efficient (Supplementary Fig. 9, and Supplementary Movie 2), some OM proteins were not exchanged. For example, TraA itself was not transferred (Fig. 5a). One explanation for this is that TraA may indirectly bind to the cell wall by interacting with TraB (Fig. 4b). Moreover, it should be noted that exchanging TraA receptors in nature could be detrimental, because cells bearing different receptors may permit promiscuous OME interactions whereby polymorphic toxins are also transferred as cargo[19].

Cell motility is indirectly required for OME[14]. As shown in Supplementary Fig. 10, co-incubation of nonmotile donors and recipients resulted in no transfer. However, incidental contacts between nonmotile cells still caused TraA foci to form,

suggesting that gliding motility is not required for this step. What role motility plays in OME remains an open question, but one hypothesis is that gliding movements between cells bound by TraA/B may stress the adhesins and membranes to trigger OM fusion. Further studies are needed to test this idea.

**TraA/B assembles myxobacteria into a multicellular tissue.** Myxobacteria represent intriguing examples of how single-celled organisms transition into a cooperative multicellular lifeform. We sought to assess how myxobacteria make this transition by using our suite of OME reporters. First, cells expressing TraA-FPs were incubated at a high density that mimics natural swarm conditions. This resulted in many foci being formed between cells, indicating individuals assembled into a cohesive population through TraA/B contacts (Fig. 5b and Supplementary Fig. 11), and supports our prior conclusion that cargo is serially transferred between cells[19]. These contacts link cells together in a network that resemble gap junctions found in eukaryotic tissue where cells are held together and cellular content is also exchanged[24]. In addition, natural myxobacterial populations are mobile and live and adapt to different microenvironments. Consequently, when these populations merge, their proteomes

and physiological states are likely different, and as such incongruences between populations can lead to disharmony[25,26]. We hypothesize that OME enables neighboring cells to form intercellular junctions that facilitate communication and cellular exchange that, in principle, helps cells to establish homeostasis in a population. To simulate this, we labeled two kin populations with GFP and mCherry cargo to represent different adaptations and mixed them together. At the initial time point, the populations were clearly distinct representing two different adaptation states (Fig. 5c and Supplementary Fig. 12). Strikingly, after 90 min of OME this was no longer the case and the two cell types could no longer be distinguished; the populations became phenotypically homogeneous. In contrast, when isogenic populations containing incompatible *traA* alleles were mixed, there was no OME, and the populations, therefore, remained distinct. These observations demonstrate how a cell surface receptor governs kin recognition, which leads to intercellular junction formation and content exchange that facilitates the transition of individual myxobacteria into cooperative tissue-like structures (see Supplementary Movie 3 for a working model).

## Discussion

In this work, we uncovered new insights into how social bacteria recognize kin and assemble cells into a cohesive tissue-like population. Although kin-directed behavior is known in several microbial systems[1,2], our knowledge of the underlying mechanisms remains limited, in part due to the challenge in real-time visualization of cell–cell recognition at the molecular level. Here, by directly tagging proteins involved in social recognition and behavior, we show that myxobacteria identify kin through a molecular 'handshake,' followed by the exchange of their private cellular goods. To our knowledge, this is the first report to directly visualize kin recognition at the molecular level. Our work provides mechanistic insights into how multicellularity may have arisen from unicellular life. Our findings are striking considering that the transition only relies on a single locus, i.e. the *traAB* operon. Here, the TraA/B adhesin provides a solution to stabilizing multicellularity by increasing the survival fitness of a group over free-living individuals. That is, TraA/B allows related cells to identify each other and subsequently engage in beneficial interactions through sharing their cellular goods and establishing homeostasis in a population[17,20]. To further ensure the success of cooperation by many individuals in a multicellular entity, OME also contains a second discrimination step to verify that the engaging cells are true clonemates[19]. Notably, a large suite of polymorphic toxins are transferred by OME[19]. Consequently, interactions among myxobacteria that fortuitously contain compatible TraA receptors, but different repertoires of toxins and cognate immunity proteins, lead to lethal outcomes for cells that are not recent descendants because the immunity proteins are not transferred during OME. Thus, TraA recognition in combination with toxin-mediated discrimination provides an effective means to protect multicellular cooperation from being exploited by non-kin or cheaters. Last, cellular exchange may allow the physiological information of cells, contained within their OM proteome and lipidome, to be shared within a population. This exchange likely helps to establish homeostasis and in turn paves the way for tissue formation and multicellularity (Fig. 5d). By analogy, cell–cell recognition and content exchange also occurs in sperm–egg interactions during fertilization. This cooperative process typically initiates multicellularity in plants and animals where a new individual starts from a single fused cell. Our study provides mechanistic clues for how cell–cell recognition and content exchange, mediated by a cell surface adhesin, facilitates the emergence of multicellularity.

On a separate but interesting topic, we show that OM proteins in *M. xanthus* are surprisingly fluid by the use of a suite of OM fluorescent reporters. This finding contrasts with the prevailing view that Gram-negative bacteria, based on studies from *E. coli*, harbor a rigid OM wherein proteins and lipopolysaccharides (LPS) are generally thought to be immobile[21,23,27]. Our novel observations can, however, be explained by known differences in the cell envelop between these organisms. First, *E. coli* cells resemble rigid rods, which in part is attributed to the properties and organization of molecules in their OM[23]. In contrast, *M. xanthus* cells are less rigid and are analogous to the flexibility of a wet noodle (e.g., Supplementary Fig. 5). This flexibility contributes to their fluid gliding movements and other functions. Second, the OM of *E. coli* and other enterics serve as an exceptional barrier that excludes noxious molecules (e.g., hydrophobic antibiotics and detergents) from entering the cell[28], while myxobacteria do not[29]. The barrier properties of the *E. coli* OM is protected by quality control systems that remove aberrant phospholipids accumulated in the outer leaflet that compromises the OM barrier[30,31]. In contrast, *M. xanthus* lacks these systems and thus phospholipids likely accumulate in their outer leaflet, which may contribute toward their sensitivity to hydrophobic compounds and toward their OM membrane fluidity. Finally, some myxobacteria lack LPS and instead contain alternative lipids in the OM[32] that are less bulky and likely more fluid. Although *M. xanthus* has LPS, it also contains these other lipids that presumably reside in the OM[33]. Taken together, we suggest that differences in the OM composition between *E. coli* and *M. xanthus* impact the fluidity of their OMs. These differences likely reflect how these species evolved. For example, *E. coli* adapted to the mammalian intestinal tract where bile detergents are in high concentration, while *M. xanthus* did not, and instead it requires a flexible cell envelope for gliding motility.

Interestingly, we found that different types of OM proteins appear to vary in their diffusion kinetics (Supplementary Fig. 13). That is, TraA and TraB diffuse at a slower rate compared to SS$_{OM}$-GFP. The relatively rapid diffusion of SS$_{OM}$-GFP is likely attributed to its lipoprotein nature and its subcellular localization on the inner leaflet of the OM. We suspect that the rapid diffusion of OME cargo ensures its efficient lateral transfer through transient fusion junctions between cells. However, it should be noted that the mobility of myxobacterial OM proteins reported here is considerably lower than that of the inner membrane proteins or cytoplasmic proteins[34,35]. This finding is broadly in line with the mentioned work that the OM is an important load-bearing structure that provide stiffness and strength of cells[23]. Nevertheless, our work provides new insights into OM dynamics and organization in gram-negative bacteria. We hypothesize that the fluidity of myxobacteria OMs is an important feature that facilitates individual cells to transition towards a multicellular lifestyle.

## Methods

**Bacterial strains and growth conditions**. Strains used in this study are listed in Supplementary Table 1. *M. xanthus* cells were cultured in CTT medium (1% [w/v] casitone; 10 mM Tris-HCl, pH 7.6; 1 mM KH$_2$PO$_4$; 8 mM MgSO$_4$) at 33 °C with shaking. To prepare plates, 1.5% (w/v) agar was added to CTT. To prepare agarose pads for microscopy, casitone was reduced to 0.2% (w/v), and 1.2% (w/v) agarose was added to the medium. *E. coli* strains were cultured in LB medium. When necessary, 50 μg ml$^{-1}$ of kanamycin (Km), 300 μg ml$^{-1}$ streptomycin (Sm), or 50 μg ml$^{-1}$ zeocin (Zeo) was added to the medium.

**Plasmid and strain construction**. Plasmids and primers used in this study are listed in Supplementary Table 1 and Supplementary Table 2. To create TraA-FP fusions, GFP (a monomeric version of superfolder GFP) or mCherry was inserted between C2 and C3 repeats of TraA. Briefly, the *pilA* promoter (P$_{pilA}$), *traA* fragments, and *gfp* or *mCherry* were first PCR amplified. The resulting amplicons that contain ~25 bp overlaps between adjacent fragments were ligated into pDP22

vector (linearized with EcoRI and XbaI) in Gibson Assembly Master Mix (New England Biolabs). When necessary, *traB* was also cloned downstream of *traA-mCherry* of *traA-gfp* in the pDP22 vector. pPC42 (*traA ΔVD-gfp-traB*) was constructed in a similar manner as described above, except that primers were designed to remove the VD from TraA. To create pPC48, the fragment of P*pilA*-*traA-gfp* from pPC47 (digested with EcoRI and HindIII) was sub-cloned into pXW7 (linearized with EcoRI and HindIII). To create TraB-GFP fusion, the fragments of P*pilA*, *traB*, and *gfp* were PCR amplified and ligated into pDP22 (linearized with EcoRI and XbaI) through Gibson Assembly. If needed, *traA* was also cloned upstream of *traB-gfp* in the pDP22 vector. The preceding TraA-FP and TraB-FP fusions were generated using TraA/B from *M. xanthus* DK1622. The TraA[Mf]-FP fusions (pPC45-46) were constructed in a similar manner as described above, except that the fragments of *traA[Mf]* were PCR amplified. To create pPC43, *mCherry* in pXW6 was swapped with *gfp* through Gibson Assembly. In brief, P*pilA* with SS$_{OM}$ was first PCR amplified using pXW6 as a template. The resulting fragment, and *gfp*, were then ligated into pXW6 (linearized with EcoRI and HindIII).

All plasmids were verified by PCR, restriction enzyme digestion, and if necessary, DNA sequencing. Verified plasmids were then electroporated into *M. xanthus* cells and selected with appropriate antibiotics. All pSWU19 and pKSAT derived plasmids were integrated into the genome at the Mx8 attachment site, whereas pPC47 was integrated at the Mx9 attachment site. The markerless in-frame deletion of *traAB* in DK10410 was made as described[8]. In brief, pDP35, containing a *traAB* deletion cassette and a Km$^r$-*galK* selection cassette, was first electroporated into DK10410 and integrated at the *traAB* loci by homologous recombination and Km$^r$ selection. The plasmid was then excised by counter-selection on 1% galactose plates. The resulting markerless deletion of *traAB* was verified by PCR with flanking primers.

**Microscopy.** Generally, cells were mounted on glass slides or agarose pads and observed under a Nikon E800 phase contrast/fluorescence microscope equipped with Texas Red and FITC filter sets and ×100 and ×60 oil objective lenses. Some microscopy was done with a Zeiss Axio Imager Z2 epifluorescence microscope equipped with a Hamamatsu Orca-Flash4.0 sCMOS camera, 46HE and 63HE filter sets, and a ×100 oil objective lens. Time-lapse microscopy was done with live cells mounted on an agarose pad using the Zeiss microscope equipped with automated autofocus. Time series were processed with ZEN imaging software (Zeiss). FRAP assays were performed with confocal illumination provided by an LMM5 laser launch (Spectral Applied Research). Confocal images were acquired on an Olympus IX-81 microscope equipped with a ×100 oil objective lens, a Yokogawa spinning disk CSU-X1, and an ORCA-Flash 4.0 sCMOS camera. The supply and integration of all imaging components were carried out by Biovision Technologies.

**Protein transfer assay.** Transfer assays were essentially done as described[13]. In brief, to test for SS$_{OM}$-mCherry or SS$_{OM}$-GFP transfer, bacterial strains were first grown to mid-log phase (density of ~5 × 10$^8$ cells per ml). Cells, without washing, were directly mixed in desired combinations. Mixtures of cells were spotted on agarose pads and incubated at 33 °C for at least 30 min before imaging. To record the time-lapse transfer of SS$_{OM}$-GFP, imaging was started immediately after the cell mixtures was dried on the agarose pads.

**Stimulation assay.** This assay was done as described[8] to test the functions of TraB-GFP. Briefly, *M. xanthus* cells were grown to mid-log phase in CTT. Cells were then washed and resuspended in TPM buffer (10 mM Tris-HCl, pH 7.6; 1 mM KH$_2$PO$_4$; 8 mM MgSO$_4$) to a calculated density of 2.5 × 10$^9$ cells per ml. A Δ*traB* donor strain expressing *traB-gfp* was mixed with a recipient strain expressing native *traAB* at 1:1 ratio. Mixtures were spotted onto 1/2 CTT (CTT medium with 0.5% Casitone) agar plates supplemented with 2 mM CaCl$_2$. After overnight incubation at 33 °C, the colony edges were imaged.

**FRAP.** *M. xanthus* strains expressing TraA-GFP, TraB-GFP, or SS$_{OM}$-GFP were grown to mid-log phase, and cells were directly mounted on thin agarose pads. FRAP was carried out with the confocal microscope as noted above. The iLas$^2$ system was used to control laser power and position. Laser was pulsed at selected regions of the cell for 3−5 times to achieve sufficient photobleaching. One pre-bleach image and a series of post-bleach images were acquired using Metamorph 7.7 software (Molecular Devices). Fluorescence recovery was observed in TraA-GFP (*n* = 51), TraB-GFP (*n* = 24), and SS$_{OM}$-GFP (*n* = 11) cells. FRAP experiments of which the image series were acquired over long timescale (~5 min for TraA-GFP or TraB-GFP, ~2.5 min for SS$_{OM}$-GFP) were then selected for quantitative analyses (Supplementary Fig. 13).

The fluidity of OM reporters was assessed by quantifying the fluorescence recovery within bleached areas. To correct for fluorescence loss due to image acquisition, the relative fluorescence recovery was measured by calculating the ratio of fluorescence intensity of the bleached area to that of the whole cell at each time point. In addition, kymographs of the one-dimensional fluorescence intensity profiles of cells before and after photobleaching are shown. Briefly, fluorescence intensity profiles along the long axes of cells were first determined using ImageJ software[36] and then normalized to the mean intensity in each frame to correct for

the intensity loss during image acquisitions. The normalized values were used to produce an intensity heatmap.

Besides measuring the relative fluorescence recovery as described above, we also normalized the fluorescence intensities of the bleached area (*I*$_{bleach}$) and the whole cell area (*I*$_{total}$) to compare the recovery kinetics between different experiments (see Supplementary Fig. 13), using the following equations[37],

$$I_{(t)} = \frac{I_{FRAP(t)}}{I_{FRAP(0)}} \times \frac{I_{total(0)}}{I_{total(t)}} \tag{1}$$

$$I_{norm} = \frac{I_{(t)} - I_{(t_0)}}{I_{(0)} - I_{(t_0)}} \tag{2}$$

where 0 is pre-bleach, and $t_0$ is immediately after photobleaching. The normalized intensities of bleached areas (*I*$_{norm}$) were fitted to the following equations to determine halftime of recovery and mobile fraction,

$$Y_{(t)} = A \times \left(1 - e^{\frac{\ln(0.5)}{t_{half}} \times t}\right) \tag{3}$$

where A is the mobile fraction, and $t_{half}$ is the halftime of recovery.

Diffusion coefficient (*D*) of SS$_{OM}$-GFP was determined as described[38]. Briefly, the one-dimensional fluorescence profiles of the long axis of cells were measured and normalized as described above. Postbleach profiles were then subtracted from the prebleach profile to generate a series of difference profiles, which approximately follow a Gaussian shape. The profiles were fitted to a Gaussian curve to generate *C* (the peak height of the curve) and *R* (half-width of the curve). *D* was obtained by plotting the bleach depth *C* versus time *t* according to the following equation,

$$C = C_0 R_0 \left(R_0^2 + 8Dt\right)^{-0.5} \tag{4}$$

where $C_0$ and $R_0$ are the values extracted from the first postbleach image.

**Antibody binding assay.** This assay was done with a fluorescence microscope equipped with a custom flow cell. As depicted in Supplementary Fig. 3A, live cells were immobilized on a poly-L-lysine−coated coverslip. The coverslip was then secured to a glass slide with double-sided tape. The resulting chamber was filled with TPM buffer and buffer containing appropriate antibodies was then flowed through the chamber. Solutions were infused from one side of the chamber and a Kimwipe was applied to the other side to generate capillary flow. Primary anti-TraA serum[3] was used at a 1:500 dilution, and a horseradish peroxidase (HRP)-conjugated goat anti-rabbit secondary antibody (Pierce, cat.nr. 31460) was used at a 1:250 dilution. Antibody solutions were incubated for > 30 min at room temperature. After primary antibody incubation, immobilized cells were washed 8−10 times with TPM. Cells were imaged before and after antibody treatment.

**Protease accessibility assay.** This assay was done as described[8] with minor changes. Briefly, cells were grown to mid-log phase, washed with TPM, and resuspended in TPM containing different concentrations of PK (New England Biolabs) for 10 min at 33 °C with agitation. To stop the reaction, complete EDTA-free protease inhibitor cocktail (Roche) was added at a 1 × final concentration. Next, cells were washed with TPM containing 0.1 × protease inhibitor cocktail and boiled for 15 min to fully inactivate PK. SDS sample buffer was then added to prepare whole-cell lysates for western blot analysis.

PK accessibility of fluorescent fusion proteins was analyzed by fluorescence microscopy. First, live cells were immobilized onto poly-L-lysine−coated coverslip of the flow cell chamber as described above. The chamber was then filled with TPM buffer containing 100 μg ml$^{-1}$ PK and incubated for at least 5 min. Fluorescence images of cells before and after PK treatment were acquired. PK accessibility of protein targets was determined by measuring the reduction in fluorescence intensity after PK treatment.

**Immunofluorescence.** This assay was essentially done as described[8]. Briefly, cells were grown to mid-log phase and fixed with 1.6% paraformaldehyde and 0.025% glutaraldehyde for 10 min. After washing with PBS buffer (137 mM NaCl, 2.7 mM KCl, 10 mM Na$_2$HPO$_4$, 1.8 mM KH$_2$PO$_4$; pH 7.4), cells were immobilized on a poly-L-lysine−coated diagnostic slide. Next, cells were blocked in PBS containing 2% BSA and subsequently incubated with primary anti-TraA serum[3] (1:3000 dilution) and secondary Alexa Fluor 594−conjugated donkey anti-rabbit IgG (Jackson ImmunoResearch, cat.nr. 711-585-152; 1:150 dilution) before imaging. All incubations were done at room temperature.

**Western blot.** TraA western blot analyses were done as described[8]. To generate anti-TraB serum, two TraB peptides, LNRRVDFTIQPPSDGPRP and KRPDRVFSGVSVGDM, were synthesized as antigens. Pre-immune sera from five rabbits were screened by western blot analyses against *M. xanthus* whole-cell lysates to select the rabbits exhibiting minimal background cross-reactivity for immunization (Thermo Scientific Pierce Protein Biology). Primary anti-TraA serum[3] was used at a 1:40,000 dilution, anti-TraB serum at a 1:15,000 dilution, and anti-GFP antibody (Invitrogen, cat.nr. A11122) at a 1:20,000 dilution. An HRP-conjugated goat anti-rabbit secondary antibody (Pierce, cat.nr. 31460) was used at a 1:15,000 dilution.

**Plasmolysis**. This assay was essentially done as described[14] with a few changes. Here a flow cell was used to track plasmolysis of specific cells. Briefly, live cells were immobilized onto poly-L-lysine−coated coverslips within a flow cell chamber filled with TPM as described above. The chamber was then filled with 0.5 M NaCl to induce plasmolysis. Cells were imaged before and after the treatment.

**Filamentation assay**. To induce cell filamentation, 30 µg ml$^{-1}$ of cephalexin (GoldBio) was added to cell cultures at early log phase and then incubated for >4 h. Cells were mounted on agarose pads and imaged by fluorescence microscopy.

**Fluorescence image analysis**. Fluorescence intensity profiles of cells were analyzed with ImageJ software[36]. For direct comparisons between TraA-GFP and TraA-mCherry fluorescence intensity profiles, the peak fluorescence signal was first normalized for each profile to 1. The remaining fluorescence profiles were then normalized based on their signal correlation with the peak signal. The normalized intensities were plotted in arbitrary units. To quantify cargo transfer, fluorescence profiles of donors ($F_{donor}$) and recipients ($F_{recipient}$) were first measured. To correct for fluorescence loss during image acquisition, the relative fluorescence of donors and recipients was assessed by calculating the ratio of $F_{donor}$ or $F_{recipient}$ to the total fluorescence ($F_{donor} + F_{recipient}$) at each time point. To generate demographs in Fig. 4e, the fluorescence profiles were analyzed as described above and then processed in R version 3.5.1 (http://www.r-project.org)[39] using a Cell Profiles script (https://github.com/ta-cameron/Cell-Profiles)[40].

**Statistics**. All experiments were performed at least three times with similar results. The exact $n$ values for protease accessibility assay, FRAP assay, and antibody binding assay are provided in the figure legends or in the corresponding methods section. For the remaining microscopy experiments, >100 cells or cell pairs were analyzed with similar observations, and representative images are shown. For Fig. 1c, unpaired and two-tailed $t$-tests were performed to analyze the significance differences between groups.

**Reporting summary**. Further information on research design is available in the Nature Research Reporting Summary linked to this article.

## Data availability
The data generated during this study are provided within the manuscript or the Supplementary Information files. The source data underlying Figs. 1c, 2b–d, 3a, b, 4c, d and Supplementary Figs 1b, 2a–c, 4b, 6c, 8c, d, 9a, b and 13a–f are provided as a Source Data file. Additional information that supports the findings of this study is available from the corresponding author upon request.

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

## Acknowledgements
We thank the Bowman and Gatlin labs for assistance and use of microscopes. We thank Chris Vassallo and other members of the Wall lab for helpful suggestions. This work was supported by the National Institutes of Health grant GM101449 to D.W.

## Author contributions
P.C. and D.W. designed experiments, analyzed data, and wrote the paper. P.C. contributed new reagents/tools and performed experiments.

## Additional information

**Competing interests:** The authors declare no competing interests.

