## [Peer Review File · Nature Communications]

Reviewers' comments:

Reviewer #1 (Remarks to the Author):

This paper reports on the mechanism by which *Myxococcus xanthus* cells engage in kin recognition that ultimately leads to exchange of materials such as toxins. The basis for recognition is a family of polymorphic cell surface receptors (TraA proteins) that self-associate in the membrane. TraA, presumably a lipoprotein, is aided by a beta-barrel outer membrane protein, TraB. The authors propose that a fluid outer membrane is the basis for the TraA proteins forming clusters as cells touch each other in a kin-specific manner. This is an interesting story that follows recent work from this and other laboratories. The authors should address the following points.

Major Points

Membrane fluidity. This is a critical part of the mechanism proposed but is barely discussed in the paper. The authors show FRAP curves for functional GFP fusions of both TraA (Fig. 2) and TraB (Fig. S5), both of which show recovery within minutes. These are striking data since recent evidence (cited in the paper; refs 20-22) suggests proteins in the outer membrane of another Gram-negative bacterium, *E. coli*, have highly restricted mobility and do not show such behaviour. The authors do not discuss the implications of their observations yet this is at the heart of the mobility they observe. A number of issues should be addressed.

- (1) Is the outer membrane of *M. xanthus* similar to that of *E. coli* in terms of composition?
- (2) Has the mobility of other lipoproteins in the outer membranes of Gram-negative bacteria been similarly shown to be mobile? If so, some comparison is warranted.
- (3) Did the authors extract diffusion coefficients from the recovery curves? The recoveries they observe appear slow relative to inner membrane proteins, for example, where recovery is complete within seconds.
- (4) On p5, line 100, the authors say that 'TraA-mCherry formed foci in <30 s upon cell-cell contact' yet FRAP data suggest a timescale of at least 1-2 minutes. Are these compatible observations?
- (5) Can the authors be certain that what they observe is fluidity as opposed to an active process driven the proton-motive force, as is gliding motility itself?
- (6) The TraB recovery curve (Fig. S5) is particularly intriguing since this is not only a beta-barrel protein (which ref. 20 showed are typically immobile in the outer membrane) but also contains a peptidoglycan binding domain similar to OmpA. OmpA has also been shown to be immobile by FRAP (Verhoeven et al (2013) BMC Microbial. 13, 66).
- (7) The TraB knockout did not form TraA foci (Fig. 4), but is TraA mobility affected, e.g. in a FRAP experiment? This would be informative.
- (8) No error bars are shown for the FRAP data. These should be shown along with the number of replicates in each dataset.

Minor Points

1. Very little information is given about the proteins at the centre of this work. It was not obvious from the outset that TraA is a lipoprotein, for example, nor how big it is (no. of amino acids). Similarly, SSom-GFP/mCherry. No explanation was given as to what this protein is and hence why it being mobile/exchangeable made sense.
2. Fig. S2. It was not clear why clustering of TraA-GFP was not induced simply by the primary antibody, which is bivalent. Shouldn't this also induce clustering?

Reviewer #2 (Remarks to the Author):

This well-written and interesting study from the Wall laboratory provides the first (visual) evidence that the cell surface receptor of *M. xanthus*, TraA, is highly mobile on the cell surface. Notably,

upon contact with neighboring cells with a compatible traA allele, the receptors appear to cluster at contact sites based on a fluorescent reporter within TraA. This is the main new result provided in this study, which goes on to show that exchange depends on compatible traA alleles and on TraB, and that TraB co-localizes with the TraA clusters at sites of contact. Notably, FRAP analysis indicated that TraB is also fluid in membranes, which seems counterintuitive if it is connected to the cell wall as depicted in Fig. 4B. Like any innovative study, this one raises many new questions, including: when a fluorescent TraA signal is visualized between cells, is it primarily due to restriction in movement of interacting TraAs, TraA clustering, or both? That is, can you assume you are visualizing clustering of receptors, or could fluorescence be due to restriction in mobility? What is the role of the cell wall binding domain of TraB in these interactions, and in exchange of cellular goods?

Comments:

1. It is notable that TraA exchange, but not outer membrane exchange, occurs between non-motile cells (Fig. S9). Clearly there is more to membrane exchange than the TraA "handshake" alone. Did the authors also visualize TraB under these conditions to see if co-localization still occurred under non-motile conditions, or if the fluorescent signals elicited by TraA/TraB are altered under these conditions?

2. p. 6, lines 105-107. Antibodies against the variable domain of TraA caused TraA-GFP fluorescence to occur, interpreted as receptor clustering. The conclusion is that this data supports the hypothesis that TraA is fluid in the OM. As a control, did the authors also carry this out using F(ab) monovalent ab fragments, in solution or immobilized on a nanobead? This would address the possibility that fluorescence may occur due to restriction in movement of TraA, and not necessarily due to TraA clustering.

David Low

Reviewer #3 (Remarks to the Author):

Dr. Wall's group has made seminal contributions on kin recognition and outer membrane exchange (OME) of myxobacteria. However, TraA, the key player of OME, was only visualized by immunofluorescence. This manuscript is a long-awaited report that shows the dynamics of TraA and the correlation between TraA clusters and OME. Overall, this is an excellent report. The experiments are well designed and executed, and the manuscript is written in a logic and concise manner.

I have a few minor comments:

1, TraA-GFP and TraA-mCherry seem to be over-expressed in all the experiments. This needs to be stated clearly in the main text. What does TraA look like when expressed under its native promoter? A supplementary figure will help to justify the over-expression.

2, Fig. S2. The authors used antibodies to show the fluidity of TraA. In my opinion, the FRAP experiments are sufficient for this purpose. To me, the antibody-binding experiment suggests that cell-cell contact is not required for the formation of TraA clusters. Instead, any cell-surface factors that either slow down TraA diffusion and/or promote TraA aggregation will trigger the formation of TraA clusters. If this is the case, it might be a good idea to move the the antibody-binding experiment to the next section "A molecular handshake governs kin recognition in myxobacteria". With better explanation, this experiment could be used to elucidate the mechanism by which TraA-TraA binding between contacting cells promotes the formation of TraA clusters (Fig. 2E). Moreover, this experiment could also explain why TraA clusters are not formed 100% along cell-cell junctions (Cao & wall, 2017, PNAS). Following this logic, maybe Fig. 3A-C, Fig. S2 and Fig. 2E could be reorganized into a new figure.

3, the FRAP experiments in Fig. 2. Panels 2C and 2D suggest that while diffusing TraA molecules continue to join the clusters, the molecules in the clusters never escape. If this is the case, one would expect to see that when two cells maintain contact long enough, such as in Fig. 3B, all the fluorescence signal will be in the clusters. Does this happen in reality?

4, the discussion about "bacterial tissue" seems a little over stretched. To me, Fig. 5B is just a repeat of Fig. 3, only at a larger scale. It is still a question how much force the intercellular TraA-TraA interaction can provide, to connect individual cells into a tissue. I do believe that OME plays roles in cell repair and the maintenance of colony homogeneity. However, since only outer membrane contents are exchanged, I am not so sure about how much "physiological information" is shared through OME.

We thank the reviewers for their constructive comments. Our responses are in blue italics below. Corresponding changes in the revised manuscript are highlighted therein.

Reviewers' comments:

Reviewer #1 (Remarks to the Author):

This paper reports on the mechanism by which *Myxococcus xanthus* cells engage in kin recognition that ultimately leads to exchange of materials such as toxins. The basis for recognition is a family of polymorphic cell surface receptors (TraA proteins) that self-associate in the membrane. TraA, presumably a lipoprotein, *(TraA does not contain a lipobox but it does contain a type I signal sequence. Please see Minor Point 1 below for more details)* is aided by a beta-barrel outer membrane protein, TraB. The authors propose that a fluid outer membrane is the basis for the TraA proteins forming clusters as cells touch each other in a kin-specific manner. This is an interesting story that follows recent work from this and other laboratories. The authors should address the following points.

Major Points

Membrane fluidity. This is a critical part of the mechanism proposed but is barely discussed in the paper. The authors show FRAP curves for functional GFP fusions of both TraA (Fig. 2) and TraB (Fig. S5), both of which show recovery within minutes. These are striking data since recent evidence (cited in the paper; refs 20-22) suggests proteins in the outer membrane of another Gram-negative bacterium, *E. coli*, have highly restricted mobility and do not show such behaviour. The authors do not discuss the implications of their observations yet this is at the heart of the mobility they observe. A number of issues should be addressed. *We agree and two paragraphs addressing this topic were added to the Discussion.*

(1) Is the outer membrane of *M. xanthus* similar to that of *E. coli* in terms of composition? *The OMs of E. coli and M. xanthus are both asymmetric and contain phospholipids in the inner leaflet and LPS in the outer leaflet. However, there are important differences. First, M. xanthus is sensitive to very low levels of various types of detergents, which is in stark contrast to E. coli, which, for example, grows in the presence of 10% SDS! Thus in M. xanthus, and likely other myxobacteria, the OM serves as a poor barrier, which is consistent with their genomes lacking the quality control pathways found in E. coli that remove aberrant phospholipids from the outer leaflet. Second, some myxobacteria, e.g. *Sorangium* spp, completely lack LPS and instead contain sphingolipids and related lipids. M. xanthus also produces these lipids and likely has them in the OM. Finally, E. coli cells are very rigid and the OM is thought to contribute toward that rigidity. In contrast, M. xanthus cells are very flexible and behave like a 'wet noddle,' e.g. see videos/micrographs in manuscript, suggesting fundamental differences in their OMs. These are important points and the text has been modified accordingly.*

(2) Has the mobility of other lipoproteins in the outer membranes of Gram-negative bacteria been similarly shown to be mobile? If so, some comparison is warranted. *According to this work and our prior work, we found that lipoproteins in M. xanthus are typically mobile unless they are restricted within a macromolecular complex. Moreover, mobility is an important property that allows lipoproteins to be efficiently transferred during OME. In fact, SS_{OM}-GFP used in this study is a lipoprotein where GFP is tagged with a lipoprotein signal peptide (type II) and is mobile (Figs. S9 & S13). In an earlier study in E. coli (Ghosh et al. J. Bacteriol., 2005, 187 (6)), a fraction of OM proteins were shown to be mobile using a labeling strategy, although the natures of the mobile OM proteins were not clear.*

(3) Did the authors extract diffusion coefficients from the recovery curves? The recoveries they observe appear slow relative to inner membrane proteins, for example, where recovery is complete within seconds.

In the revised manuscript, we estimated the diffusion coefficient (D) of SS_{OM}-GFP (see Fig. S9 legend). Compared to SS_{OM}-GFP, the signals of TraA/B-GFP are patchy (i.e. intrinsic heterogeneity), which is

problematic for extracting D values. Thus, we calculated their recovery halftime and mobile fraction instead (Fig. S13). In addition, we generated kymographs of the one-dimensional fluorescence profiles for the FRAP images of TraA/B-GFP and SS_{OM}-GFP (Fig. 2, Fig. 4C and Fig. S9A), to help readers better visualize their recovery kinetics. The revised Discussion also address these points.

(4) On p5, line 100, the authors say that 'TraA-mCherry formed foci in <30 s upon cell-cell contact' yet FRAP data suggest a timescale of at least 1-2 minutes. Are these compatible observations? *Typically, TraA foci form within 30 s after cell-cell contact initiates. In FRAP experiments, incidental contacts (TraA foci formation) between cells already occurred before photobleaching. After photobleaching, the bleached molecules presumably stayed at contact interfaces for some time, which likely would impede the ability of unbleached TraAs from joining these clusters. This explains why foci recovery from FRAP analysis was slower.*

(5) Can the authors be certain that what they observe is fluidity as opposed to an active process driven the proton-motive force, as is gliding motility itself? *Yes, for several reasons: i) Various myxobacterial OM components including TraA/B, SS_{OM}-GFP (a lipoprotein reporter), and OM lipids (our analysis & Ducret et al, eLife 2 (2013): e00868) were shown to be mobile using FRAP. These OM components are not linked to PMF. ii) Gliding motility in myxobacteria is driven by PMF through the AglRQS proton channel, which is a core motor component analogous to flagellar motility. However, from saturating genetic screens (Dey & Wall, J Bacteriol., 2014 & unpublished), we have not identified PMF components required for OME. Additionally, when AglRQS or gliding motility is knockout OME still occurs. iii) According to our microscopy observations, TraA/B and SS_{OM}-GFP molecules/clusters appear to move in a stochastic manner, unlike the rotating helical movement exhibited by PMF-powered gliding motility motors.*

(6) The TraB recovery curve (Fig. S5) is particularly intriguing since this is not only a beta-barrel protein (which ref. 20 showed are typically immobile in the outer membrane) but also contains a peptidoglycan binding domain similar to OmpA. OmpA has also been shown to be immobile by FRAP (Verhoeven et al (2013) BMC Microbial. 13, 66). *Yes, we agree. In contrast to previous reports in E. coli (refs 21-23 in the paper) and Pseudomonas aeruginosa (White et al. PNAS. 114.45 (2017): 12051-12056.), the discovery of TraB mobility is intriguing, and therefore we moved the TraB FRAP data to Fig. 4. Interestingly, another study on E. coli (Verhoeven et al, BMC Microbiol. 2013; 13: 66.) showed that when the peptidoglycan binding domain of OmpA was removed it was still immobile, supporting the lack of OM fluidity in E. coli. In unpublished work, when the peptidoglycan binding domain of TraB was removed it retained mobility.*

(7) The TraB knockout did not form TraA foci (Fig. 4), but is TraA mobility affected, e.g. in a FRAP experiment? This would be informative. *Good point. We found that TraA mobility was not affected in a TraB knockout background and vice versa. However, our findings of TraA/B fluidity are well-supported and this evidence does not make a significant addition to this paper.*

(8) No error bars are shown for the FRAP data. These should be shown along with the number of replicates in each dataset. *For simplicity, we only showed the recovery curves of the representative FRAP images in the main figures. In the revised manuscript, we conducted additional analyses where individual FRAP curves were normalized and plotted together in Fig. S13 (with number of replicates and error bars). We also included additional examples of TraA-GFP FRAP in Fig. S2, as a complementary data to Fig. 2.*

Minor Points

1. Very little information is given about the proteins at the centre of this work. It was not obvious from the outset that TraA is a lipoprotein, for example, nor how big it is (no. of amino acids). Similarly, SS_{OM}-GFP/mCherry. No explanation was given as to what this protein is and hence why it being mobile/exchangeable made sense. *TraA does not contain a lipobox. However, it does contain a type I*

signal peptide and a C-terminal sorting motif called MYXO-CTERM that localizes TraA onto the cell surface (ref 13 and unpublished work). We have evidence that the MYXO-CTERM is cleaved and posttranslationally modified, but the nature of the modification is unknown (note: TraA cell surface localization occurs independently of TraB). SS_{OM}-GFP/mCherry used in this study are lipoprotein reporters (see above). Additional information has been added to the revised manuscript to help readers understand the nature of TraA/B and SS_{OM}-GFP/mCherry.

2. Fig. S2. It was not clear why clustering of TraA-GFP was not induced simply by the primary antibody, which is bivalent. Shouldn't this also induce clustering? *We do see moderate clustering when only using the primary antibody (see Fig. S3, additional foci were seen at cell poles). Additionally, prolonged primary antibody incubations generally yield more foci, but we also found that adding the secondary antibody noticeably increased clustering.*

Reviewer #2 (Remarks to the Author):

This well-written and interesting study from the Wall laboratory provides the first (visual) evidence that the cell surface receptor of *M. xanthus*, TraA, is highly mobile on the cell surface. Notably, upon contact with neighboring cells with a compatible traA allele, the receptors appear to cluster at contact sites based on a fluorescent reporter within TraA. This is the main new result provided in this study, which goes on to show that exchange depends on compatible traA alleles and on TraB, and that TraB co-localizes with the TraA clusters at sites of contact. Notably, FRAP analysis indicated that TraB is also fluid in membranes, which seems counterintuitive if it is connected to the cell wall as depicted in Fig. 4B. *Yes, but we hypothesize TraB conditionally binds or slides along the cell wall. The figure/legend has been modified.* Like any innovative study, this one raises many new questions, including: when a fluorescent TraA signal is visualized between cells, is it primarily due to restriction in movement of interacting TraAs, TraA clustering, or both? That is, can you assume you are visualizing clustering of receptors, or could fluorescence be due to restriction in mobility? *Our working model is that homotypic binding between TraA receptors from opposing membranes restricts TraA mobility, which in turn allows them to accumulate at cell-cell contacts. We did not preclude the possibility that TraA clustering also plays a role. That is, TraA may exhibit multivalent binding during homotypic interactions that aggregates multiple molecules into one cluster. However, we do not think there is strong lateral interaction between TraA molecules on the same cell membrane, since (i) TraA foci do not form without cell-cell contacts and (ii) when two touching membranes separate, TraA clusters readily disassociate.* What is the role of the cell wall binding domain of TraB in these interactions, and in exchange of cellular goods? *Interestingly, our unpublished work suggests that removal of the cell wall binding domain still allows cell-cell adhesion and TraA/B clustering, but it abolishes OME. We thus hypothesize this domain facilitates membrane fusion after cell-cell adhesion is established, perhaps by transiently binding the cell wall resulting in membrane stress around the TraA/B foci as cells move past one another.*

Comments:

1. It is notable that TraA exchange, but not outer membrane exchange, occurs between non-motile cells (Fig. S9. *Now Fig. S10*). *Correction, TraA cluster formation, rather than TraA exchange, occurs between nonmotile cells. However, being competent for motility in partnering cells is not required for OME per se because nonmotile cells will exchange when a 3rd party motile cells are mixed with nonmotile strains even when those cells lack TraA (e.g., Wei et al, 2011, Mol. Micro).* Clearly there is more to membrane exchange than the TraA "handshake" alone. Did the authors also visualize TraB under these conditions to see if co-localization still occurred under non-motile conditions, or if the fluorescent signals elicited by TraA/TraB are altered under these conditions? *Based on our observations, motility per se will not cause a change in TraA/B co-localization or the TraA-TraA "handshake". We think motility plays an indirect role*

during OME. For example, as mentioned above, cell movement likely generates mechanical stress on membranes and proteins at cell-cell adhesion sites that further triggers membrane fusion to occur.

2. p. 6, lines 105-107. Antibodies against the variable domain of TraA caused TraA-GFP fluorescence to occur, interpreted as receptor clustering. The conclusion is that this data supports the hypothesis that TraA is fluid in the OM. As a control, did the authors also carry this out using F(ab) monovalent ab fragments, in solution or immobilized on a nanobead? This would address the possibility that fluorescence may occur due to restriction in movement of TraA, and not necessarily due to TraA clustering. *Interesting point. We have not conducted these experiments. In case of cell-cell contacts, one key idea is that TraA homotypic interactions will restrict the movement of TraA within a confined area (i.e. cell-cell contact sites) where they accumulate. However, if we simply put cells in monovalent ab solution, one would expect the movement of TraA molecules on the whole cell (rather than a confined area) will slow to some extent, but not cause foci.*

Reviewer #3 (Remarks to the Author):

Dr. Wall's group has made seminal contributions on kin recognition and outer membrane exchange (OME) of myxobacteria. However, TraA, the key player of OME, was only visualized by immunofluorescence. This manuscript is a long-awaited report that shows the dynamics of TraA and the correlation between TraA clusters and OME. Overall, this is an excellent report. The experiments are well designed and executed, and the manuscript is written in a logic and concise manner. *Thank you.*

I have a few minor comments:

1, TraA-GFP and TraA-mCherry seem to be over-expressed in all the experiments. This needs to be stated clearly in the main text. *This information was added to the Results section of the revised manuscript. Note that expression was still single copy from the chromosome using the heterologous P_{ilA} promoter.* What does TraA look like when expressed under its native promoter? A supplementary figure will help to justify the over-expression. *We have constructed a functional GFP fusion into the native traA locus. However, we cannot definitely detect a fluorescence signal, apparently because the wild-type expression level of TraA is very low. Similarly, it is difficult to detect native TraA expression by western analysis. Having said this, we have confirmed the functions of all the TraA/B-FPs used in this study and conducted our experiments with proper controls.*

2, Fig. S2. The authors used antibodies to show the fluidity of TraA. In my opinion, the FRAP experiments are sufficient for this purpose. To me, the antibody-binding experiment suggests that cell-cell contact is not required for the formation of TraA clusters. Instead, any cell-surface factors that either slow down TraA diffusion and/or promote TraA aggregation will trigger the formation of TraA clusters. *Mostly likely.* If this is the case, it might be a good idea to move the antibody-binding experiment to the next section "A molecular handshake governs kin recognition in myxobacteria". With better explanation, this experiment could be used to elucidate the mechanism by which TraA-TraA binding between contacting cells promotes the formation of TraA clusters (Fig. 2E). Moreover, this experiment could also explain why TraA clusters are not formed 100% along cell-cell junctions (Cao & wall, 2017, PNAS). *Good point.* Following this logic, maybe Fig. 3A-C, Fig. S2 and Fig. 2E could be reorganized into a new figure. *We agree that the antibody-binding data provides useful information for different aspects of our work. While the figure organization was left unchanged, we included the Fig. S2 (now Fig. S3) citation and additional discussion under relevant sections.*

3, the FRAP experiments in Fig. 2. Panels 2C and 2D suggest that while diffusing TraA molecules continue to join the clusters, the molecules in the clusters never escape. If this is the case, one would

expect to see that when two cells maintain contact long enough, such as in Fig. 3B, all the fluorescence signal will be in the clusters. Does this happen in reality? *We have not seen this. There is always a fluorescence signal surrounding the whole cell. We do think TraA clusters are dynamic and molecules within clusters can escape (although at a low rate). According to Fig. 2C, unbleached molecules can join the cell-cell junction that was packed with bleached molecules, suggesting TraA-TraA homotypic interactions might be transient, and preexisting molecules constrained by homotypic bindings might disassociate from the clusters and free up space for new molecules to join. Additionally, there may be a limit to the number of TraA receptors that can join a cluster. Finally, as shown in Fig. S13F, we estimate there is about a 30% immobile fraction of TraA-GFP on the cell surface, and therefore it is unlikely that all TraA-GFPs are available to cluster at cell-cell contacts.*

4, The discussion about "bacterial tissue" seems a little over stretched. To me, Fig. 5B is just a repeat of Fig. 3, only at a larger scale. It is still a question how much force the intercellular TraA-TraA interaction can provide, to connect individual cells into a tissue. *Fig. 5B shows that many cells, in fact nearly all cells, within biofilms form TraA foci, highlighting the likelihood that these cells are exchanging their cellular contents simultaneously. However, we agree that TraA-TraA interactions are not the primary 'force' holding cells together; instead this force is provided by the extracellular matrix (polysaccharides and various proteins). We have modified our model (Fig. 5D) and legend to better illustrate our point.* I do believe that OME plays roles in cell repair and the maintenance of colony homogeneity. However, since only outer membrane contents are exchanged, I am not so sure about how much "physiological information" is shared through OME. *Myxobacteria, like other gram-negative bacteria, physically interact with their environment and other cells through their OM and, therefore, the protein/lipid composition of the OM is a key indicator of how cells have adapted ('physiological information') to their local environments. Unfortunately, little is known about the composition of the M. xanthus OM and how cells adapt to changing environments. Although we currently do not have direct evidence, we propose a framework where the exchange of OM components transfers physiological information about how cells have adapted to their local environments. In the revised manuscript we have expanded this discussion and softened the tone.*

REVIEWERS' COMMENTS:

Reviewer #1 (Remarks to the Author):

The authors have address all the points I raised and all those of the other reviewers. This makes a great (and now more understandable) story.